

# Genomic comparison of non-photosynthetic plants from the family Balanophoraceae with their photosynthetic relatives

Mikhail I. Schelkunov[1,2], Maxim S. Nuraliev[3,4] and
Maria D. Logacheva[1]

[1] Skolkovo Institute of Science and Technology, Moscow, Russia
[2] Institute for Information Transmission Problems, Moscow, Russia
[3] Faculty of Biology, Moscow State University, Moscow, Russia
[4] Joint Russian–Vietnamese Tropical Scientific and Technological Center, Hanoi, Vietnam

## ABSTRACT

The plant family Balanophoraceae consists entirely of species that have lost the ability to photosynthesize. Instead, they obtain nutrients by parasitizing other plants. Recent studies have revealed that plastid genomes of Balanophoraceae exhibit a number of interesting features, one of the most prominent of those being a highly elevated AT content of nearly 90%. Additionally, the nucleotide substitution rate in the plastid genomes of Balanophoraceae is an order of magnitude greater than that of their photosynthetic relatives without signs of relaxed selection. Currently, there are no definitive explanations for these features. Given these unusual features, we hypothesised that the nuclear genomes of Balanophoraceae may also provide valuable information in regard to understanding the evolution of non-photosynthetic plants. To gain insight into these genomes, in the present study we analysed the transcriptomes of two Balanophoraceae species (*Rhopalocnemis phalloides* and *Balanophora fungosa*) and compared them to the transcriptomes of their close photosynthetic relatives (*Daenikera* sp., *Dendropemon caribaeus*, and *Malania oleifera*). Our analysis revealed that the AT content of the nuclear genes of Balanophoraceae did not markedly differ from that of the photosynthetic relatives. The nucleotide substitution rate in the genes of Balanophoraceae is, for an unknown reason, several-fold larger than in the genes of photosynthetic Santalales; however, the negative selection in Balanophoraceae is likely stronger. We observed an extensive loss of photosynthesis-related genes in the Balanophoraceae family members. Additionally, we did not observe transcripts of several genes whose products function in plastid genome repair. This implies their loss or very low expression, which may explain the increased nucleotide substitution rate and AT content of the plastid genomes.

Corresponding author
Mikhail I. Schelkunov,
shelkmike@gmail.com

# INTRODUCTION

It is commonly believed that all plants generate organic substances through photosynthesis. However, there were several dozen independent cases when a

photosynthetic plant obtained the ability to get organic substances from other plants or fungi; this is called mixotrophy or partial heterotrophy (reviewed by *Těšitel et al. (2018)*). Certain mixotrophic plants can later lose their photosynthetic ability and begin to rely solely on parasitism on their plant or fungal host, this is called complete heterotrophy, holo-heterotrophy, or just "heterotrophy" (reviewed by *Merckx, Bidartondo & Hynson (2009)* and by *Nickrent (2020)*).

This transition to heterotrophy leaves a noticeable trace on plant morphology and ecology. Heterotrophic plants generally possess either no leaves or very reduced leaves. These plants also possess little to no green colouring owing to the absence or very low levels of chlorophyll. The majority of heterotrophic species spend most of the year completely underground, as they do not directly require light for survival. Instead, they appear above-ground for reproduction, with some exceptions like *Rhizanthella gardneri* and *Hydnora triceps* which flower belowground (*Musselman & Visser, 1989*; *Thorogood, Bougoure & Hiscock, 2019*). Additionally, heterotrophic plants exhibit other morphological and ecological changes, which have been reviewed, for example, by *Leake (1994)* and *Těšitel (2016)*.

With the exception of certain rare cases (*Molina et al., 2014*; *Cai et al., 2021*), plants possess three genomes that include those contained in plastids, mitochondria, and nuclei. The plastid genome is the smallest and possesses the highest copy number, thereby alleviating sequencing and assembly (*Sakamoto & Takami, 2018*). These characteristics in combination with other features of plastid genomes are the reason why most plant genomes that have been sequenced and assembled to date are plastid (*Twyford & Ness, 2017*). As the majority of plastid genes are required for photosynthesis, it is unsurprising that one plastid genome alteration, which is characteristic of non-photosynthetic plants, is a massive plastid gene loss. Another widely observed feature is an increase in the nucleotide substitution rate; however, this is likely not owing to relaxation of natural selection. The third common feature is an increase in AT content. While the cause of the first feature is obvious, the causes underlying the second and third features remain unknown. *Wicke & Naumann (2018)* reviewed these alterations and other characteristics of non-photosynthetic plants.

The mitochondrial genomes of non-photosynthetic plants have been studied less than the plastid genomes, but, probably, a feature common to many parasitic plants is the horizontal transfer of genes from the mitochondrial genomes of their hosts to the mitochondrial genomes of the parasites (*Mower, Jain & Hepburn, 2012*; *Sanchez-Puerta et al., 2017*, *2019*; *Roulet et al., 2020*).

Sequencing of whole nuclear genomes can be expensive owing to their large size. Therefore, transcriptome sequencing may be more suitable to obtain information about genes from very large nuclear genomes. Studies of nuclear genomes, including those based on transcriptome analysis, revealed several features that are characteristic of the genomes of non-photosynthetic plants (*Wickett et al., 2011*; *Lee et al., 2016*; *Schelkunov, Penin & Logacheva, 2018*; *Yuan et al., 2018*; *Cai et al., 2021*). There is an extensive loss of photosynthesis-related genes within the nuclear genome, similar to that observed in plastid genomes. However, for a yet-unknown reason, genes that encode proteins that function

in chlorophyll synthesis are often retained. It is hypothesized that chlorophyll may possess certain functions other than those required for photosynthesis, for example it can function in photoprotection, photodetection or be a precursor during the course of synthesis of other substances (*Cummings & Welschmeyer, 1998*; *Barrett et al., 2014*). Just as in plastid genomes, for an unknown reason, the nuclear genes of non-photosynthetic plants evolve faster and without signs of relaxed selection. The AT content of nuclear genes of non-photosynthetic plants is not much increased or not increased at all, unlike the plastid genomes.

Among the most unusual plastid genomes currently known to scientists are plastid genomes of non-photosynthetic plants from the family Balanophoraceae (*Su et al., 2019*; *Schelkunov, Nuraliev & Logacheva, 2019*; *Chen et al., 2020a*). The family Balanophoraceae comprises several dozen species that inhabit tropical and subtropical areas and feed by attaching to the roots of different plants and absorbing nutrients from them (*Hansen, 2015*). An analysis based on a relaxed molecular clock model suggested that Balanophoraceae appeared approximately 110 million years ago (*Naumann et al., 2013*). Plastid genomes have been sequenced for one species of the genus *Rhopalocnemis* (*Schelkunov, Nuraliev & Logacheva, 2019*) and four species of the genus *Balanophora* (*Su et al., 2019*; *Chen et al., 2020a*). These plastid genomes are about a tenth of the size of plastid genomes of typical photosynthetic plants. The AT content in the currently known plastid genomes of Balanophoraceae is in the range of 86.8–88.4% (*Chen et al., 2020a*), making them the most AT-rich of all known plant genomes with regard to not only plastids but also mitochondrial and nuclear genomes. This large AT content affects not only non-coding regions but also genes, with AT contents of some genes exceeding 90%. The nucleotide substitution rate in the known plastid genomes of Balanophoraceae is more than 10-fold greater than that in photosynthetic relatives. Additionally, at least some plastid genomes of plants from the genus *Balanophora* have altered their genetic code, where the TAG codon now codes for tryptophan instead of being a stop codon (*Su et al., 2019*). As noted above, such large AT contents and substitution rates currently have no accepted scientific explanation.

Given the unusual features of the plastid genomes of Balanophoraceae, we decided to investigate the patterns of mutation accumulation and gene loss in the nuclear genomes of these plants to test how strongly the alterations in these two genomes are correlated and to propose possible causative links. Our previous estimate for *Rhopalocnemis phalloides* suggests a nuclear genome size of at least 30 Gbp (*Schelkunov, Nuraliev & Logacheva, 2019*); thus, nuclear genome sequencing would be expensive. Instead, we sequenced the transcriptome of *Rhopalocnemis phalloides* and also used the transcriptome of *Balanophora fungosa*, another plant from the Balanophoraceae family (Fig. 1), that was sequenced as part of the 1 KP Project (*Carpenter et al., 2019*). For comparison, we used the mixotrophic plants *Daenikera* sp., *Dendropemon caribaeus*, and *Malania oleifera*. These three mixotrophic species are members of three different families of the order Santalales that also encompasses the family Balanophoraceae. The families are, according to the APG IV classification system (*The Angiosperm Phylogeny Group, 2016*), Olacaceae, Santalaceae, and Loranthaceae. Similar to plants from the Balanophoraceae family,

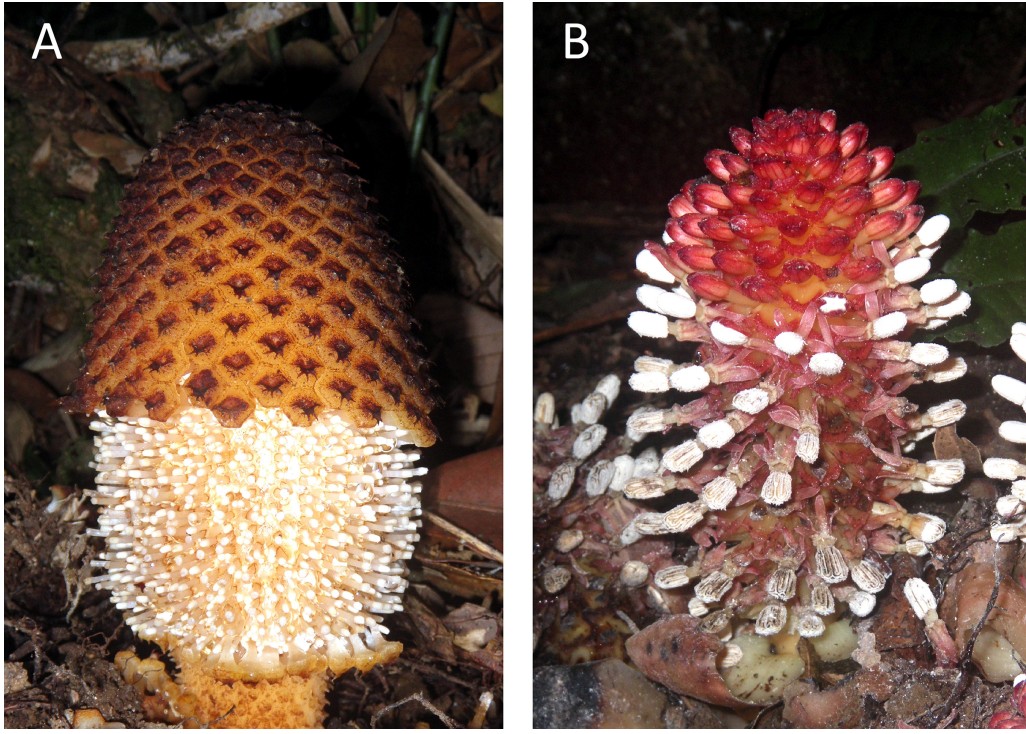

**Figure 1 Photos of the studied non-photosynthetic species.** The photos depict above-ground parts of *Rhopalocnemis phalloides* (A) and *Balanophora fungosa* (B). Photographed by M. Nuraliev.

these three mixotrophic species are capable of parasitism. However, they are also obligate autotrophs that must photosynthesize and cannot survive without light (*Vidal-Russell & Nickrent, 2008*; *Caraballo-Ortiz et al., 2017*; *Li, Mao & Li, 2019*). The transcriptomes of these three species have been previously sequenced as parts of different projects (*Xu et al., 2019*; *Carpenter et al., 2019*).

# MATERIALS AND METHODS

## Sample collection and sequencing

The sequenced specimen of *Rhopalocnemis phalloides* was collected during the expedition of the Russian-Vietnamese Tropical Centre in Kon Tum Province, Vietnam, in May 2015. The specimen was preserved in silica gel and RNAlater storage solution. The voucher was deposited at the Moscow University Herbarium (*Seregin, 2018*) under the barcode MW0755444.

We extracted RNA from the inflorescence. RNA extraction was performed using the RNEasy Mini kit (Qiagen, Venlo, the Netherlands) following the manufacturer's instructions. The only modification was the addition of Plant Isolation Aid Solution (ThermoFisher, Waltham, MA, USA) to the lysis buffer. Selection of RNA was made using the Ribo-Zero Plant Leaf kit (Illumina, San Diego, CA, USA). Further library preparation was performed with the NEBNext Ultra II RNA kit (New England Biolabs, Ipswich, MA, USA).

The library was sequenced on:

1. Illumina NextSeq 500, producing 6,912,802 paired-end 75 bp-long reads (3,456,401 read pairs).
2. Illumina HiSeq 2500, producing 54,794,466 paired-end 125 bp-long reads (27,397,233 read pairs).
3. Illumina HiSeq 4000, producing 181,190,240 paired-end 76 bp-long reads (90,595,120 read pairs).

Overall, 242,897,508 reads were produced for *Rhopalocnemis phalloides* (121,448,754 read pairs).

## Transcriptomes of comparison

For comparison, we used transcriptomes from *Balanophora fungosa*, *Daenikera* sp., *Dendropemon caribaeus*, and *Malania oleifera*. *Balanophora fungosa*, *Daenikera* sp., and *Dendropemon caribaeus* were sequenced as part of the 1KP project (*Carpenter et al., 2019*), while *Malania oleifera* was sequenced by *Xu et al. (2019)*.

The transcriptomes from *Balanophora fungosa*, *Daenikera* sp., *Dendropemon caribaeus*, and *Malania oleifera* were sequenced on an Illumina HiSeq 2000, producing:

1. For *Balanophora fungosa*, 26,470,118 paired-end 90 bp-long reads (13,235,059 read pairs). The identifier for these reads in the NCBI Sequence Read Archive (SRA) is ERR2040275. According to personal communication with Drs. Bruno Fogliani and Matthieu Villegente who collected the sample, the RNA used for sequencing was extracted from the inflorescence.
2. For *Daenikera* sp., 22,878,728 paired-end 90 bp-long reads (11,439,364 read pairs). The SRA identifier of the reads is ERR3487343. According to personal communication with Drs. Bruno Fogliani and Matthieu Villegente who collected the sample, the RNA used for sequencing was extracted from leaves.
3. For *Dendropemon caribaeus*, 22,393,054 paired-end 90 bp-long reads (11,196,527 read pairs). The SRA identifier of the reads is ERR2040277. The RNA used for sequencing was extracted from whole seedlings.
4. For *Malania oleifera*, 222,952,532 paired-end 150 bp-long reads (111,476,266 read pairs). The SRA identifiers of the reads are SRR7221530, SRR7221531, SRR7221535, SRR7221536, and SRR7221537. The RNA used for sequencing was extracted from fruits and leaves.

The use of different organs from the three photosynthetic plants may affect the results of this study. However, our analysis of the transcriptomes was qualitative and not quantitative, *i.e.*, we did not analyse expression levels, which reduces the effect of using transcriptomes from different organs.

To minimize methodological differences, we assembled the transcriptomes of these 4 species using the exact methods used for the transcriptome of *Rhopalocnemis phalloides*

(see below). For *Rhopalocnemis phalloides* and *Malania oleifera* that possess several read datasets, the datasets were combined prior to assembly.

## Read processing and transcriptome assembly

Reads were trimmed using Trimmomatic 0.39 (*Bolger, Lohse & Usadel, 2014*), performing 5 procedures in the following order:

1. Adapters were trimmed according to the palindromic method.
2. Bases possessing a Phred score of less than three were trimmed from the 3′ end.
3. If a group of four consecutive bases with an average Phred score of less than 15 existed, this group was trimmed together with the portion of the read that was in the 3′ direction from that group.
4. If the average Phred score of the read was less than 20, the read was discarded.
5. If the length of the read after the previous four steps was less than 30 bases, it was discarded.

The assembly was performed using Trinity 2.8.4 (*Haas et al., 2013*) with digital normalization to 50× coverage. The minimum contig length was set to 200 bp. The expression levels of the assembled transcripts were quantified using Salmon 0.9.0 (*Patro et al., 2017*). As the major isoform of a gene, we defined the isoform with the highest expression level measured by the "transcripts per million" (TPM) value. All other isoforms ("minor" isoforms) were discarded.

Contigs possessing a very low coverage may contain misassemblies. To detect such contigs, the reads were mapped to all contigs using CUSHAW 3.0.3 (*Liu, Popp & Schmidt, 2014*) requiring at least 80% of a read to map with a sequence similarity of at least 98%. Contigs possessing an average coverage of less than 3× were discarded.

Protein-coding sequences (CDSs) were predicted using TransDecoder 5.5.0 (*Haas et al., 2013*), using its in-built capabilities and also using the presence of Pfam-A domains in open reading frames (ORFs) and the similarity of ORF sequences to sequences within the NCBI NR database. The NCBI NR database was current as of 13 May 2019. The Pfam-A domain prediction in ORFs found by TransDecoder was conducted using the Pfam-A 32.0 database (*El-Gebali et al., 2019*) and the hmmscan tool from the HMMER 3.2 package (*Mistry et al., 2013*) with the default parameters. The similarity search between ORFs and NCBI NR sequences was performed using the "blastp" command from DIAMOND 0.9.25 (*Buchfink, Xie & Huson, 2015*) with the "–more-sensitive" option and a maximum e-value of $10^{-5}$. An ORF was considered a probable CDS if at least one of the following three criteria was met:

1. TransDecoder considered this ORF a likely CDS based on its in-built criteria such as the hexanucleotide frequencies.
2. A Pfam-A domain was found in the ORF.
3. The ORF had a match in NCBI NR.

After CDS prediction, we removed the CDSs with proteins having best matches in NCBI NR not to Embryophyta, thus removing contamination. We hereafter refer to proteins as translated CDSs. CDSs with proteins that had no matches in NCBI NR and consequently did not possess any definite taxonomic assignment were retained.

The completeness of the transcriptome assemblies was assessed by BUSCO 3.1.0 (*Simão et al., 2015*) using the eukaryotic set of conserved proteins. The eukaryotic set was preferred over plant sets, as plant sets contain a number of proteins that function in photosynthesis and should thus be absent in *Rhopalocnemis phalloides* and *Balanophora fungosa*.

## Transcriptome annotation

The transcriptome was annot ated using the following three methods:

1. Method of reciprocal best hits (RBH). The major isoforms of *Arabidopsis thaliana* proteins from the TAIR10 database (*Berardini et al., 2015*) were aligned using BLASTP 2.9.0+ (*Camacho et al., 2009*) to proteins of each of the five studied species with the maximum e-value set to $10^{-5}$, word size three, and the low-complexity filter switched off. Then, vice versa, the proteins of the five species were aligned in the same way to the proteins of *Arabidopsis thaliana*. If a pair of proteins were RBHs to each other in both of these alignments, the corresponding protein in the studied species was supposed to have the same functions as its match in *Arabidopsis*.

2. The Gene Ontology (GO) annotation (*The Gene Ontology Consortium, 2019*). Proteins of the five studied species were annotated with GO terms by PANNZER2 (*Törönen, Medlar & Holm, 2018*) using a positive predictive value threshold of 0.5. The relaxed criteria for query and subject lengths were switched on using the option "–PANZ_FILTER_PERMISSIVE".

3. KEGG metabolic annotation (*Kanehisa et al., 2016*). The annotation was performed using the GhostKOALA (*Kanehisa, Sato & Morishima, 2016*) web server on 24 December 2019.

The second and third methods were used only for the nuclear proteins of the studied species. To remove transcripts of mitochondrial and plastid proteins, we removed all transcripts whose proteins were RBHs to mitochondrial and plastid proteins of *Arabidopsis thaliana* in the first analysis.

## Phylogenetic tree and the natural selection analysis

To construct the phylogenetic tree of the five studied species, we determined the orthogroups of their CDSs and the CDSs of *Arabidopsis thaliana*. The CDSs of *Arabidopsis thaliana* used for this analysis were CDSs from the major transcript isoforms from the TAIR10 database. To restrict the analysis to nuclear CDSs, mitochondrial and plastid CDSs were excluded as described above. Orthology was determined by OrthoFinder 2.3.8 (*Emms & Kelly, 2019*) using DIAMOND as a tool for similarity calculation.

Next, to build the tree we used all orthogroups that had exactly one sequence from each species, there were 1,039 such orthogroups. They were aligned using TranslatorX

1.1 (*Abascal, Zardoya & Telford, 2010*) with MAFFT 7.402 (*Katoh & Standley, 2013*). The source code of TranslatorX was altered to use MAFFT in the E-INS-i mode to allow for large gaps in the alignment. Large gaps may arise in alignment if orthologous transcripts from different species contain different exons. The poorly aligned regions of the orthogroups were removed using Gblocks 0.91b (*Castresana, 2000*) with default parameters. Then, the alignments for different orthogroups were concatenated into one alignment. The tree for this alignment was built using RAxML 8.2.12 (*Stamatakis, 2014*) with the GTR+Gamma model using 20 starting trees and with the number of bootstrap pseudoreplicates automatically determined by the autoMRE method. The GTR+Gamma model was used, because it is one of the most general substitution models. The GTR +Gamma+I is also frequently used; however, the primary developer of RAxML advises against using this model (*Stamatakis, 2016*).

The selection analysis was performed by PAML 4.9 (*Yang, 2007*) using the branch model with the $F3 \times 4$ codon frequencies model, a starting dN/dS of 0.5, and a starting transition/transversion ratio of 2. The option "cleandata" that removes columns with gaps and stop codons was switched on. The tree given to PAML was the tree produced by RAxML. The number of substitutions that occurred on tree branches was calculated using PAML. To calculate 95% confidence intervals for the dN/dS values of branches, we generated 1,000 bootstrap pseudoreplicates for the alignment, performed 1,000 separate PAML calculations, and then used the 2.5 and 97.5 percentiles for dN/dS on each branch.

The tree was drawn using TreeGraph 2.14.0 (*Stöver & Müller, 2010*) with *Arabidopsis thaliana* used as the outgroup. For the number of substitutions on the branches, we used the values provided by PAML and not those provided by RAxML. The AT content of nuclear genes was calculated from the same concatenated gene alignment that was used for the phylogenetic tree construction and the dN/dS evaluation. dN/dS values for the *NTH1* gene were computed using the same method as that used for the dN/dS values of concatenated genes by using the topology inferred from the concatenated genes.

## Other analyses

The GO enrichment analysis was performed only for nuclear genes, excluding plastid and mitochondrial genes as described above. For each GO term, we calculated the *p*-value for the difference in the proportion of genes that code for this GO term between each of the two non-photosynthetic species and three photosynthetic species, thus achieving six comparisons overall. The *p*-values were calculated according to Fisher's exact test using the program GOAtools 0.6.10 (*Klopfenstein et al., 2018*). We then performed the Bonferroni correction for these six comparisons and then the Benjamini–Hochberg correction for all GO terms. Differences between photosynthetic and non-photosynthetic species that exhibited *q*-values that were less than or equal to 0.05 were considered statistically significant.

Information regarding the plastid gene content in *Rhopalocnemis phalloides* and *Balanophora fungosa* was obtained from papers that described their plastid genomes (*Schelkunov, Nuraliev & Logacheva, 2019*; *Chen et al., 2020a*). The plastid genome sequences of *Daenikera* sp. and *Dendropemon caribaeus* remain unknown. However, given
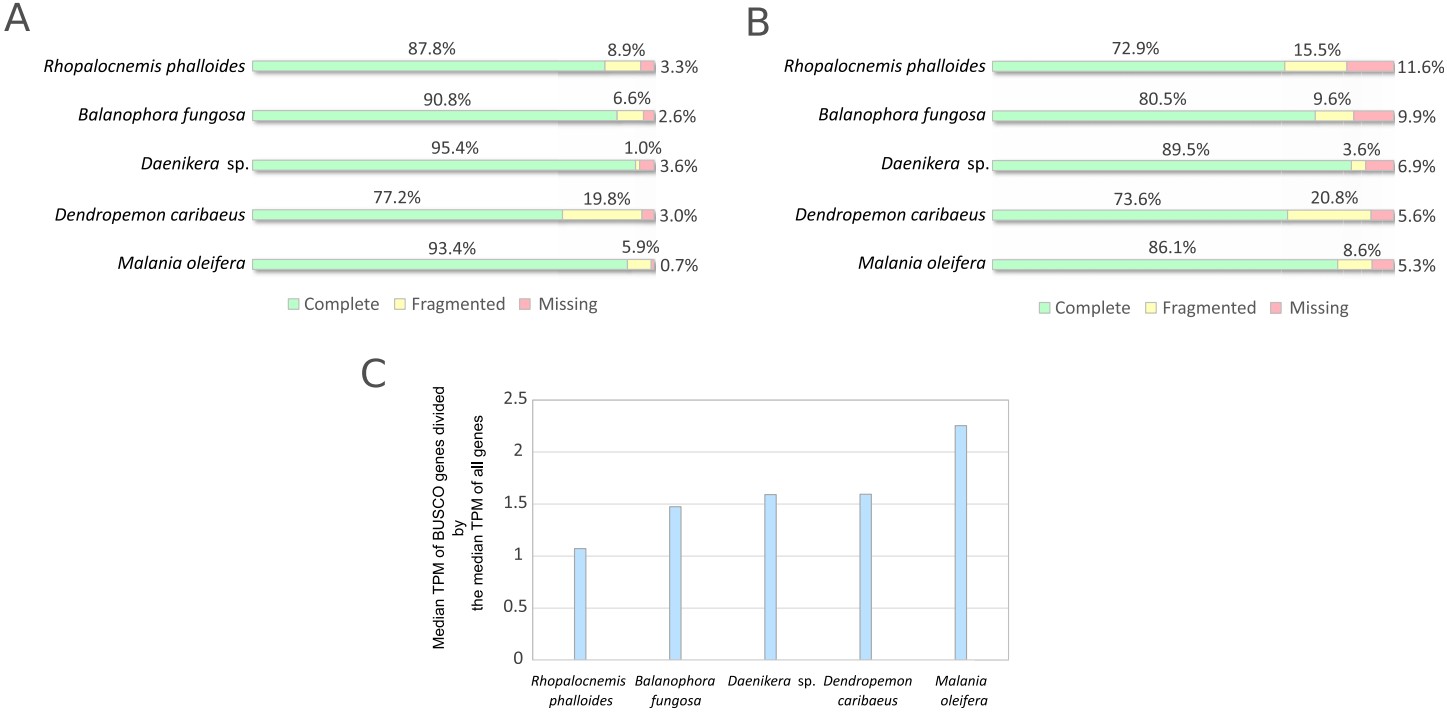

**Figure 2 BUSCO analysis results.** (A) BUSCO results for all sequences produced by Trinity. (B) BUSCO results for CDSs from major isoforms of transcripts after removal of transcripts with low coverage and contamination. (C) Comparison of expression of BUSCO genes and all genes.

the similar gene content in the plastid genome of photosynthetic Santalales (*Chen et al., 2020a*), it is reasonable to assume that the plastid gene content in *Daenikera* sp. and *Dendropemon caribaeus* would be approximately the same. There are two scientific reports examining the plastid genome of *Malania oleifera*, and these reports highly contradict each other (*Yang & He, 2019*; *Xu et al., 2019*). One of them states that the genome length is 158,163 bp, while the other describes a 125,050 bp genome. Given that the gene content in the second genome differs markedly from the gene content in other photosynthetic Santalales (*Chen et al., 2020a*), we speculate that the second genome either contains assembly mistakes or belongs to some other species that was misidentified as *Malania oleifera*. Therefore, we consider the longer genome to be the true plastid genome of *Malania oleifera*.

## RESULTS AND DISCUSSION

### The assemblies

The results of the BUSCO analysis of the assemblies are provided in Fig. 2. The results for all transcripts indicated the near absence of missing genes (Fig. 2A). However, at the stage of minor isoform removal, the percentage of missing and fragmented BUSCO genes increased (Fig. 2B). This is likely owing to our characterization of "the major isoforms" as the most expressed isoforms, and these most expressed isoforms may not possess the same exons as those of the BUSCO gene models. Such a difference in exonic content may underestimate the completeness of the BUSCO analysis. More mechanistic measures such

**Table 1 GO terms most underrepresented among nuclear genes of non-photosynthetic Santalales compared to photosynthetic Santalales.** Values in the cells are the number of genes with this GO term divided by the total number of genes with GO terms in this species.

| GO term | *Rhopalocnemis phalloides* | *Balanophora fungosa* | *Daenikera* sp. | *Dendropemon caribaeus* | *Malania oleifera* |
|---|---|---|---|---|---|
| photosystem I | 1/22583 | 0/12178 | 28/16391 | 23/22802 | 50/21755 |
| photosystem II | 1/22583 | 2/12178 | 41/16391 | 36/22802 | 62/21755 |
| photosystem | 3/22583 | 3/12178 | 57/16391 | 54/22802 | 86/21755 |
| chlorophyll metabolic process | 2/22583 | 2/12178 | 31/16391 | 32/22802 | 27/21755 |
| photosynthesis | 9/22583 | 6/12178 | 94/16391 | 98/22802 | 133/21755 |
| photosynthesis, light reaction | 7/22583 | 4/12178 | 32/16391 | 51/22802 | 61/21755 |
| thylakoid part | 19/22583 | 17/12178 | 118/16391 | 145/22802 | 172/21755 |
| photosynthetic membrane | 17/22583 | 16/12178 | 106/16391 | 129/22802 | 160/21755 |
| thylakoid | 17/22583 | 22/12178 | 125/16391 | 154/22802 | 189/21755 |
| thylakoid membrane | 14/22583 | 14/12178 | 87/16391 | 109/22802 | 130/21755 |

as N50 (the length of the longest contig such that it and all contigs longer than it constitute at least 50% of the total length of the assembly) and the total number of contigs in the assemblies are provided in Table S1.

It should be noted that BUSCO analysis may be less suitable for assessing the completeness of transcriptome assemblies as compared to that of genome assemblies if BUSCO genes are expressed at higher levels than those of average genes. A comparison of the median expression level of BUSCO genes to the median expression level of all genes (Fig. 2C) suggests that the median expression level of BUSCO genes is approximately 1.5-fold higher. In this comparison, as "all genes" we used the same CDS set as in Fig. 2B, but omitted CDSs that possessed no GO terms to reduce the number of false-positively predicted genes. This difference in expression between BUSCO genes and all genes suggests that BUSCO genes may be assembled slightly better, thus the completeness of the assembly may be overestimated.

## Nuclear gene content as inferred from the transcriptome assemblies

The results of the GO enrichment analysis indicate that all of the most statistically significant reductions in the gene sets of non-photosynthetic plants are unambiguously linked to the loss of photosynthesis.

The lists of the 10 GO terms that were most underrepresented and most overrepresented in non-photosynthetic plants compared to those in photosynthetic plants are provided in Tables 1 and 2. The complete table listing all GO terms with significantly different amounts of genes is provided in Table S2.

As shown in Table 1, photosynthesis-related genes were almost absent from the non-photosynthetic species. The GO annotation process is known to produce a number of false-positive results (*Zhou et al., 2019*), and therefore, the real reduction in the number of photosynthesis-related genes is likely stronger than that indicated in the table.

The results presented in Table 2 demonstrate that the non-photosynthetic species possess increased proportions of genes with the GO term "DNA integration" and a

**Table 2  GO terms most overrepresented among nuclear genes in non-photosynthetic Santalales compared to photosynthetic Santalales.**
Values in the cells are the number of genes with this GO term divided by the total number of genes with GO terms in this species.

| GO term | Rhopalocnemis phalloides | Balanophora fungosa | Daenikera sp. | Dendropemon caribaeus | Malania oleifera |
|---|---|---|---|---|---|
| DNA integration | 9486/22583 | 2001/12178 | 475/16391 | 344/22802 | 1714/21755 |
| DNA metabolic process | 10265/22583 | 2470/12178 | 968/16391 | 1095/22802 | 2648/21755 |
| nucleic acid metabolic process | 11374/22583 | 3449/12178 | 2266/16391 | 2967/22802 | 4308/21755 |
| nucleobase-containing compound metabolic process | 11604/22583 | 3617/12178 | 2534/16391 | 3368/22802 | 4643/21755 |
| heterocycle metabolic process | 11720/22583 | 3731/12178 | 2717/16391 | 3583/22802 | 4830/21755 |
| cellular aromatic compound metabolic process | 11735/22583 | 3733/12178 | 2745/16391 | 3619/22802 | 4881/21755 |
| organic cyclic compound metabolic process | 11783/22583 | 3777/12178 | 2799/16391 | 3694/22802 | 4942/21755 |
| nucleic acid binding | 8244/22583 | 2887/12178 | 2000/16391 | 2819/22802 | 3591/21755 |
| cellular nitrogen compound metabolic process | 11904/22583 | 3868/12178 | 2939/16391 | 3872/22802 | 5105/21755 |
| cellular macromolecule metabolic process | 12483/22583 | 4221/12178 | 3922/16391 | 5510/22802 | 6150/21755 |

number of nucleotide metabolism-related GO terms. A BLAST analysis of genes with these GO terms indicated that they predominantly belonged to transposons. Enrichment by these GO terms was previously observed in non-photosynthetic plants of the genera *Epipogium* and *Hypopitys* (*Schelkunov, Penin & Logacheva, 2018*). We cannot provide a clear explanation for this phenomenon. It can be hypothesized that this is a sign of transposon expansion within the genomes of non-photosynthetic plants. However, genomic data regarding non-photosynthetic plants are still insufficient to state this with confidence. A study of the nuclear genome of a non-photosynthetic plant *Sapria himalayana* indicates expansion of transposons in this species, thus supporting our hypothesis (*Cai et al., 2021*). Of the five species studied in our work, there are two nuclear genome size estimates that included a genome of greater than 30 Gb for *Rhopalocnemis phalloides* (*Schelkunov, Nuraliev & Logacheva, 2019*) and a 221 Mb genome for *Santalum album* (*Mahesh et al., 2018*). As the number of transposons within plant genomes is proportional to genome size (*Novák et al., 2020*), the number of transposons may indeed be larger in *Rhopalocnemis phalloides* as compared to that in *Santalum album*.

As an alternative method to assess changes in the gene content, we composed a list of *Arabidopsis thaliana* proteins which function in plastids and searched for RBHs in the CDS sets of the five studied transcriptomes (Table S3). This analysis indicated a vast reduction in genes linked to photosynthesis. Certain genes that encode products participating in the Calvin cycle are retained; however, these genes also exert functions beyond the Calvin cycle (*Schelkunov, Penin & Logacheva, 2018*). A feature that should be noted is the presence in non-photosynthetic species of several genes that encode products required for import into thylakoids. It is likely that *Rhopalocnemis phalloides* and *Balanophora fungosa* still possess thylakoids that have functions unrelated to photosynthesis or that the products of these genes have functions beyond import into thylakoids.

An unexpected feature of many non-photosynthetic plants is the presence of some levels of chlorophyll (*Cummings & Welschmeyer, 1998*). It was hypothesised that chlorophyll may have some functions other than photosynthesis (*Cummings & Welschmeyer, 1998*; *Barrett et al., 2014*). Analyses of transcriptomes and genomes revealed that the pathways for synthesis and breakdown of chlorophyll are indeed probably retained in some non-photosynthetic species (*Wickett et al., 2011*; *Schelkunov, Penin & Logacheva, 2018*; *Marcin et al., 2020*); however, they are likely lost in some other species (*Ng et al., 2018*; *Schelkunov, Penin & Logacheva, 2018*; *Chen et al., 2020b*). In *Rhopalocnemis phalloides* and *Balanophora fungosa*, RBH analysis indicated the complete disappearance of chlorophyll synthesis and breakdown genes.

The plastid genomes of *Rhopalocnemis phalloides* and *Balanophora fungosa* were previously shown to be reduced approximately 10-fold as compared to plastid genomes of their close photosynthetic relatives (*Schelkunov, Nuraliev & Logacheva, 2019*; *Chen et al., 2020a*). The number of genes in these two non-photosynthetic species is also reduced approximately tenfold (*Schelkunov, Nuraliev & Logacheva, 2019*; *Chen et al., 2020a*). The RBH analysis revealed no transcripts of the lost genes. This may indicate that they were not transferred to the nuclear or mitochondrial genome and instead disappeared completely.

In addition to GO enrichment and RBH analysis, we used KEGG metabolic annotation as the third method for the analysis of gene loss. Visual inspection of pathway maps produced by KEGG suggests, in addition to the conclusions that followed from the GO enrichment and RBH, that *Rhopalocnemis phalloides* and *Balanophora fungosa* lack transcripts for a number of genes that encode products participating in circadian rhythm organization (Fig. S1), terpenoid synthesis (Figs. S2 and S3), and carotenoid synthesis (Fig. S4). These changes are likely linked to the loss of photosynthesis.

## AT content in nuclear genes

It was previously demonstrated that plastid genomes of Balanophoraceae are highly AT-rich, with AT contents of the currently sequenced genomes being in the range of 86.8–88.4% (*Su et al., 2019*; *Schelkunov, Nuraliev & Logacheva, 2019*; *Chen et al., 2020a*). This makes them the most AT-rich of any plant genomes, and one of the most AT-rich among all genomes. The AT contents of their relatives from various photosynthetic families of the same order Santalales are much lower and exist in the range of 61.8–65.13%. The increased AT content of Balanophoraceae is a feature of both non-coding and coding regions, with average weighted by gene length AT contents of plastid protein-coding genes in species of Balanophoraceae being 88.1–91.3% and in photosynthetic Santalales being 60.9–63.7%. The cause of such high AT contents in the plastid genomes of Balanophoraceae remains unknown. This increase represents a long-known but still unexplained feature of the plastid genomes of non-photosynthetic plants, where Balanophoraceae is the most extreme case (*Wicke et al., 2013*; *Schelkunov et al., 2015*; *Lam, Soto Gomez & Graham, 2015*; *Bellot & Renner, 2015*; *Naumann et al., 2016*; *Lim et al., 2016*; *Logacheva et al., 2016*; *Roquet et al., 2016*; *Braukmann et al., 2017*; *Park, Suh & Kim, 2020*).

**Table 3  AT contents in nuclear genes of Santalales.**

| Species | AT content |
| --- | --- |
| *Rhopalocnemis phalloides* | 48.3% |
| *Balanophora fungosa* | 54.6% |
| *Daenikera* sp. | 54.7% |
| *Dendropemon caribaeus* | 52.8% |
| *Malania oleifera* | 54.5% |

To compare the characteristics of nuclear and plastid genomes, we analysed AT contents of nuclear genes. Although the plastid genome has only been determined for *Malania oleifera* and not for *Daenikera* sp. and *Dendropemon caribaeus*, the AT content of the plastid protein-coding genes of *Daenikera* sp. and *Dendropemon caribaeus* are probably somewhere within the range of 60.9–63.7% or close to that range, because this range is based on 29 plastid genomes of photosynthetic Santalales known to date. As presented in Table 3, the AT contents of the nuclear protein-coding genes of *Rhopalocnemis phalloides* and *Balanophora fungosa* do not differ much from those of their photosynthetic relatives. The AT content of *Rhopalocnemis phalloides* is even the lowest among the species. All differences in AT contents are statistically significant ($p$-values calculated by Fisher's exact tests $\leq 0.05$ after a Bonferroni correction) with the exception of three differences that included *Balanophora fungosa* compared to *Daenikera* sp., *Balanophora fungosa* compared to *Malania oleifera*, and *Malania oleifera* compared to *Daenikera* sp.

## Nucleotide substitution rate and selection in nuclear genes

Among the unusual features shown previously for the plastid genomes of *Rhopalocnemis* and *Balanophora* was their high nucleotide substitution rates that are approximately one order of magnitude greater than those of photosynthetic relatives from the order Santalales. One may assume that this is due to relaxed natural selection. However, it has been demonstrated that dN/dS values in the plastid genomes of *Rhopalocnemis* and *Balanophora* are significantly lower than those in their photosynthetic relatives (*Su et al., 2019*; *Schelkunov, Nuraliev & Logacheva, 2019*). Therefore, negative selection is likely stronger in *Rhopalocnemis* and *Balanophora*. Similar to the observed increase in the AT content, the increase in the substitution rate without relaxation of negative selection is supposed to be a typical feature of highly reduced plastid genomes of non-photosynthetic plants, not only the ones from the family Balanophoraceae (*Logacheva, Schelkunov & Penin, 2011*; *Schelkunov et al., 2015*; *Lam, Soto Gomez & Graham, 2015*; *Logacheva et al., 2016*; *Roquet et al., 2016*; *Braukmann et al., 2017*). In less reduced plastid genomes that still experience loss of photosynthesis-related genes, selection acting on the genes being lost is relaxed (*Barrett & Davis, 2012*; *Wicke et al., 2013*). We must note that while the calculation of the AT content and its comparison between species is straightforward, the calculation of substitution rates and selection is less reliable. The primary reason for this is the saturation on long branches which may lead to underestimation of substitution
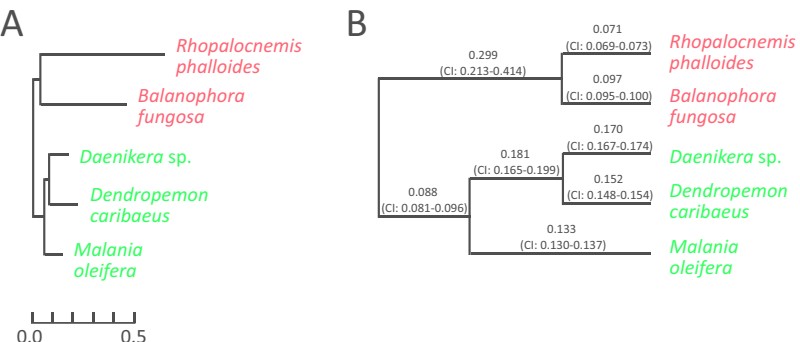

**Figure 3 Evolutionary parameters of nuclear genes in the studied Santalales.** (A) The phylogram where branch lengths represent numbers of substitutions per position. (B) The cladogram that depicts dN/dS values on branches with their 95% confidence intervals (CIs). *Arabidopsis thaliana*, used as the outgroup, is not shown. Names of non-photosynthetic species are in red, while names of photosynthetic species are in green. All bootstrap support values for the topology are 100%.

rates and underestimation of dN/dS (*dos Reis & Yang, 2013*). The underestimation of dN/dS on long branches follows from the fact that non-synonymous positions reach saturation faster than synonymous positions. Therefore, the observed lack of increased dN/dS values within the plastid genomes of non-photosynthetic species may represent a computational artefact that is a consequence of the increased substitution rate.

The analysis for nuclear genes revealed that in nuclear genes of *Rhopalocnemis phalloides* and *Balanophora fungosa*, just as in plastid ones, substitutions occur faster than in genes of photosynthetic relatives (Fig. 3A). Concurrently, the dN/dS values for nuclear genes of *Rhopalocnemis phalloides* and *Balanophora fungosa* were lower than those of the photosynthetic relatives (Fig. 3B, *p*-value < $10^{-5}$ by the likelihood ratio test). This implies stronger negative selection. Alternatively, despite the low *p*-value, this may be a consequence of the computational problem described above.

The dN/dS ratio on the branch of the common ancestor of *Rhopalocnemis phalloides* and *Balanophora fungosa* is also significantly higher (*p*-value < $10^{-5}$ by the likelihood ratio test) than that in photosynthetic plants. We are unaware of the cause of this increase in dN/dS. Although there was likely a massive loss of photosynthesis-related genes on that branch that was accompanied by relaxation of selection in those genes, the analysis of dN/dS was performed using only extant genes, thus the relaxation of selection in the lost genes cannot explain this effect.

## The causes of unusual AT contents and substitution rates in the plastid genomes of *Rhopalocnemis phalloides* and *Balanophora fungosa*

As described above, the plastid genomes of *Rhopalocnemis phalloides* and *Balanophora fungosa* exhibited increased AT contents and substitution rates. Considering that all genes that encode proteins which function in plastid genome replication, recombination, and repair (RRR) are encoded in the nuclear genome, the clue for these features of the plastid genome must be sought for in the nuclear genome.
Based on a literature analysis, we composed a list of 6 RRR proteins that were previously experimentally shown to function in RRR of the plastid genome, but not of mitochondrial or nuclear genomes (Table S3, part "Genes encoding other replication, recombination, and repair proteins that function solely in plastids"). The RBH analysis of photosynthetic species indicated that transcripts of five of these proteins were present in *Daenikera* sp., five were present in *Dendropemon caribaeus*, and six were present in *Malania oleifera*. However, we found transcripts of only one protein in *Rhopalocnemis phalloides* and only two in *Balanophora fungosa*. The transcripts absent in both *Rhopalocnemis phalloides* and *Balanophora fungosa* that were present in photosynthetic species are of the following proteins:

1. MUTS2. The exact function of this protein in plants is unknown; however, its bacterial homologs have been demonstrated to promote recombination suppression (*Pinto et al., 2005*).
2. OSB2. This protein also likely suppresses recombination (*Zaegel et al., 2006*).
3. RECA. This protein functions in recombination-dependent DNA repair (*Rowan, Oldenburg & Bendich, 2010*).

Additionally, *Rhopalocnemis phalloides* lacks the transcript of the following protein:

4. ARP. A nuclease which repairs DNA lesions resulting from oxidative damage (*Akishev et al., 2016*).

The disappearance of plastid RRR proteins gives a possible explanation for why the plastid mutation accumulation rate is highly elevated in *Rhopalocnemis phalloides* and *Balanophora fungosa*. As gene conversion, a recombination-dependent process, is supposed to be GC-biased in plastids (*Wu & Chaw, 2015*; *Niu et al., 2017*), disruption of recombination-dependent repair may cause the genome to become more AT-rich, and this may explain the high AT contents of the plastid genomes of *Rhopalocnemis phalloides* and *Balanophora fungosa*.

A transcriptomic analysis of non-photosynthetic plants of the genera *Epipogium* and *Hypopitys*, which also possess AT-rich plastids with an increased rate of nucleotide substitutions, also revealed the absence of RECA, while MUTS2, OSB2, and ARP were present (*Schelkunov, Penin & Logacheva, 2018*). This may suggest the universality of the link between the high plastid AT content, high plastid substitution rates, and loss of genes coding for RRR proteins.

An important question is why these genes are being lost in non-photosynthetic plants. The Accelerated Junk Removal (AJR) hypothesis previously proposed by our group (*Schelkunov, Nuraliev & Logacheva, 2019*) indicates that the increased mutation rate in the plastid genome may be beneficial for non-photosynthetic plants, as it accelerates the removal of pseudogenes in the plastid genome. Indeed, after a plant loses the ability to photosynthesise, selection no longer acts on photosynthesis-related genes. They begin to accumulate deleterious mutations, and their products may become dangerous to plastids. Therefore, an increase in the mutation accumulation rate may be beneficial, as it

accelerates the complete disappearance of a gene or at least the disappearance of genomic elements such as promoters of ribosomal binding sites required for gene expression. Although scientists generally study the rate of nucleotide substitutions in plastids of non-photosynthetic plants, the rates of accumulation of indels and structural mutations have also been observed to increase under these conditions (*Wicke et al., 2016*; *Wicke & Naumann, 2018*).

Analyses of substitution rates in plastid genomes from different lineages of non-photosynthetic plants indicated that the substitution rate was increased not only on the branch where photosynthesis was lost but also on the branches of descendants (*Schelkunov et al., 2015*; *Feng et al., 2016*; *Braukmann et al., 2017*; *Schelkunov, Nuraliev & Logacheva, 2019*; *Chen et al., 2020a*). If the AJR hypothesis is correct, the explanation for this may be the irreversibility of RRR gene loss. The mutation accumulation rate will be high until new RRR genes encoding plastid-targeted products evolve to replace the lost ones.

An alternative to AJR may be a hypothesis that postulates that the harm from a loss of an RRR gene is roughly proportional to the number of genes in which the mutation accumulation rate will increase after the RRR gene is lost. For example, imagine a 100,000 bp-long genome and a protein which fixes 100 errors after each replication of this genome. If the genome shortens to 10,000 bp, then after each replication this protein will fix only 10 errors. Therefore, the existence of this protein becomes less beneficial for the organism, even if strong negative selection acts on genes remaining in this small genome. This implies that when the plastid genome of a non-photosynthetic plant loses photosynthesis-related genes, the selection acting on the plastid genome RRR machinery relaxes, leading to the loss of RRR genes. This, in turn, may lead to an increase in the substitution rate and AT content of the remaining plastid genes. We further call this the Less Important Accuracy (LIA) hypothesis. A similar explanation for the negative correlation between genome sizes and their mutation rates was proposed earlier by *Drake et al. (1998)*.

Of the aforementioned 6 RRR proteins that function in plastids, a transcript of only one protein was found in both *Rhopalocnemis phalloides* and *Balanophora fungosa*. This transcript was of NTH1, a protein which functions in the base excision repair (*Roldán-Arjona et al., 2000*). Its dN/dS on the branches of the studied non-photosynthetic plants is 0.20, while on the branches of the studied photosynthetic plants it is 0.22 with a p-value for the difference (as calculated by the likelihood ratio test) of 0.92. A similar analysis conducted previously for retained RRR proteins that function in the plastids of *Epipogium* and *Hypopitys* also demonstrated that the dN/dS values of their genes do not differ significantly from those of photosynthetic relatives (*Schelkunov, Penin & Logacheva, 2018*). Thus, the selection on those RRR proteins that have survived does not appear to be relaxed.

We speculate that the increased substitution rate of the nuclear genome (Fig. 3) may have several explanations.

1. The coevolution of a host and its parasite is often described as an "arms race" in which the host constantly invents mechanisms of defence, while the parasite in turn develops means to avoid the defence. In such a situation, it may be beneficial for both the host and the parasite to increase their mutation rates (*Haraguchi & Sasaki, 1996*). As *Rhopalocnemis phalloides* and *Balanophora fungosa* rely on parasitism more than their mixotrophic relatives *Daenikera* sp., *Dendropemon caribaeus*, and *Malania oleifera* that were used as photosynthetic species for comparison, it is logical that the arms race is more intense in *Rhopalocnemis phalloides* and *Balanophora fungosa*. Thus, the higher mutation rate and, consequently, the higher rate of substitutions (fixed point mutations) may theoretically be beneficial for *Rhopalocnemis phalloides* and *Balanophora fungosa*. However, this effect can be used to explain the nuclear substitution rate but not the plastid substitution rate, as the plastid genome lacks genes whose products participate in the parasite-host interaction.

2. The AJR hypothesis can also explain the increased substitution rate in the nuclear genome, as nuclear genomes of photosynthetic plants contain many genes that encode products that function in photosynthesis (see Table S3, for example). Thus, the loss of photosynthesis and the subsequent degradation of these proteins is dangerous.

3. The LIA hypothesis may also partially explain the increase in nuclear substitution rate. However, only a small fraction of the nuclear genes of photosynthetic plants encode proteins that function in photosynthesis (Table 1). Consequently, the LIA effect is unlikely to be strong enough to explain the approximately three-fold increase in the substitution rates in the nuclear genes of *Rhopalocnemis phalloides* and *Balanophora fungosa*.

It has been reported that the substitution rate is also increased in the mitochondrial genomes of parasitic plants (*Bromham, Cowman & Lanfear, 2013*). This may be explained by the observation that there are RRR proteins common between the mitochondrial and plastid genomes, such as RECA2 (*Shedge et al., 2007*). Thus, the mitochondrial genome may also suffer if such proteins are lost or become less effective.

## CONCLUSIONS

The transcriptomic analysis of two non-photosynthetic plants from the family Balanophoraceae suggests the cause of the extreme features of their plastid genomes. The probable cause for this is the loss of nuclear-encoded proteins that function in plastid genome repair. However, *why* these proteins were lost remains enigmatic. The Accelerated Junk Removal hypothesis and the Less Important Accuracy hypothesis are two possible explanations.

The validity of these hypotheses may be tested by analysing genomes of plants that have lost photosynthesis recently. If deleterious mutations in plastid genes that are being lost accumulate with rates faster than neutral, then the Accelerated Junk Removal hypothesis is probably correct. If there are no signs of the Accelerated Junk Removal, but

one sees that the relaxation of selection acting on plastid genome repair genes encoded in the nucleus is correlated with the shortening of the plastid genome, then the Less Important Accuracy hypothesis may be correct.

## ACKNOWLEDGEMENTS

Sequencing on the HiSeq 4000 was performed using the resources of the Skoltech Genomics Core facility.

### Funding
The work was funded by the Russian Foundation for Basic Research grant No. 16-34-01003 and the budgetary subsidy to IITP RAS No. 0053-2019-0005. The work of Maxim Nuraliev was carried out as part of the Scientific Project of the State Order of the Government of Russian Federation to Lomonosov Moscow State University No. 121032500082-2. The funders had no role in study design, data collection and analysis, decision to publish, or preparation of the manuscript.

### Grant Disclosures
The following grant information was disclosed by the authors:
Russian Foundation for Basic Research: 16-34-01003 and 0053-2019-0005.
Lomonosov Moscow State University: 121032500082-2.

### Competing Interests
The authors declare that they have no competing interests.

### Author Contributions
- Mikhail I. Schelkunov conceived and designed the experiments, performed the experiments, analyzed the data, prepared figures and/or tables, authored or reviewed drafts of the paper, and approved the final draft.
- Maxim S. Nuraliev performed the experiments, prepared figures and/or tables, authored or reviewed drafts of the paper, and approved the final draft.
- Maria D. Logacheva performed the experiments, authored or reviewed drafts of the paper, and approved the final draft.

### DNA Deposition
The following information was supplied regarding the deposition of DNA sequences:

The sequencing reads of *Rhopalocnemis phalloides* are available in the NCBI Sequence Read Archive: PRJNA495456.

The transcriptome assemblies for all five studied species are available at figshare: Schelkunov, Mikhail (2021): Transcriptome assemblies for five species of Santalales. figshare. Dataset. DOI 10.6084/m9.figshare.15088440.v1.

## Data Availability

The scripts written by the authors and used in this work are available at figshare: Schelkunov, Mikhail (2020): Scripts used to study transcriptomes of Santalales. figshare. Software. DOI 10.6084/m9.figshare.13049777.v1.

## Supplemental Information

Supplemental information for this article can be found online at http://dx.doi.org/10.7717/peerj.12106#supplemental-information.

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
