# Peer review of "Genomic comparison of non-photosynthetic plants from the family Balanophoraceae with their photosynthetic relatives"

_PeerJ, doi:10.7717/peerj.12106_

## Round 0.1 · original submission · Major Revisions

I have received two evaluation reports on the submitted manuscript. Despite the effort made in data collection and laboratory, the reviewers mention drawbacks and limitations, raising some misgivings about the way it has been written up. They provided constructive comments on how the manuscript can be improved. Furthermore, I included some comments that should be considered. I hope that you will find all advice helpful when revising the manuscript.

(1) The introduction should be better reframed and authors should include some clear hypotheses.

(2) More details about the methods should be informed: (i) as it is a comparative study, information about how the RNA was extracted for all plant species should be informed; (ii) clarify why the GTR+Gamma model was used in the RaxML (line 235); (iii) considering that you applied the Bonferroni correction, I think the alfa value should be 0.0083 instead of 0.05 (line 260).

(3) The authors should improve the quality of the tables and figures presented, trying to distinguish the two kinds of plant species as you did in Figure 3.

Reviewer 1 ·

Basic reporting

The paper presents an interesting and innovative information for Balanophoraceae, showing relevance to be published by the PeerJ. However, even though the study has a lot potential, significative amount of changes and corrections are required in order to improve understanding and flow.

The authors use a somewhat informal terminology throughout the manuscript, sometimes implying personal opinion: for example, in the abstract we can see “a tremendous AT content” and “we supposed that it would be interesting”. I would advise to keep a clear and professional English language.

The background presented in the introduction could be improved. A better explanation of the family and the species investigated in the study could be added. The statement used in Line 95 does not justify the choice of the organisms investigated:
“We supposed that nuclear genomes of plants with such unusual plastid genomes also deserve being studied.”

Moreover, the text is full of strong statements that have no citation as support. I would advise to re-write the introduction using the literature as support, removing any personal opinions, and “tone down” general statements.

Experimental design

The study presents an original primary research. However, the research question could be better described (better written) to point out its relevance and how the research fills a knowledge gap.

I appreciate the effort of the authors, however the M&M section could be re-written in order to facilitate understanding, please follow examples such as Mahmood et al. (2020) (https://doi.org/10.1038/s41598-020-70406-2).

Validity of the findings

The study has potential, however I can not fully state about the methods and the validity of the findings, since there is a great amount of strong statements that are not supported by any literature throughout the manuscript (some speculations, some results from other studies without citation). In addition, the language is confusing which does not help interpretation.

Additional comments

General comments:

The paper presents interesting and innovative information for the Balanophoraceae family, showing relevance to be published by PeerJ. However, significative amount of changes and corrections are required in order to improve understanding and flow.

There is a great amount of strong statements that are not support by any literature in both the introduction and discussion sections. Another concern that I have is about the language used in the manuscript. The authors use a somewhat informal terminology throughout the manuscript, sometimes implying personal opinion: for example, in the abstract we can see “a tremendous AT content” and “we supposed that it would be interesting”. I would advise them to re-write the study and keep a clear and professional English language.

The material and methods section could also be reorganized and re-written. It is also necessary to say that there is a significant amount of “result-like” sentences within the M&M.

I present below a point-by-point comment to help in the correction process:

Abstract:
General comments:
Line 20: the word tremendous imply personal interpretation, please keep formal language. Maybe changing it by “high” or something similar.

Line 21: greater by “one order of magnitude” instead of “an order”

Line 22: Be careful with general statements, no one can possibly exhaust the literature, since there are so many papers published every day. So, it would be more acceptable to say: “To our knowledge, none of these features have yet been investigated” instead of “All these features have no definite explanations.”

Line 24: Please keep the formal language, please re-phrase “we supposed that it would be interesting”, same for Line 31 “is for an unknown reason several times larger than”, in this case, language in unprecise, one could say: “X amount”, or “X percent”, or “an average of X% higher”. The abstract should present more data, not only: “several genes”, “several times larger”

In addition: The conclusion could be improved in the abstract. What are the biological implications? What are the main “take home messages” of the study?

Introduction:
Besides informal language, the authors do not respect the format for citation. Please correct it.
• Since informal and ambiguous language was present throughout the whole manuscript, I marked the sentences that need to be changed in yellow in the pdf file.

Consider change one of the two: “this is called” on lines 41 and 43

I would advise to “tone down” general statements, and also use citations to support your statements. For example:
• Was the entire first paragraph based on Leake (1994) and Tesitel (2016)?
• Second paragraph: “This is the reason why most plant genomes sequenced and assembled to date are plastid ones.” Please be careful.

In addition, the authors say: “they do need light for survival”, however heterotrophic species do not need to be exposed to light for their survival, but they need light, since their host needs light.

Line 63 to 66: paragraph containing only one sentence. Keep formal scientific language. Same for Line 191 to 192, and phrases from lines 245 to 251.

Line 67 to 68: Personal opinion. In addition, I understand the position of the authors, however it will depend on the features chosen for sequencing. One can choose to have a high depth sequencing of a plastid and it will increase the sequencing cost.

There is no connection between the phrase on line 67 to 68 to the following sentence. In addition, transcriptome sequencing can also be expensive, sometimes more expensive than genome sequencing.

Lines 72 to 78: no literature to support strong statements and results. Same for lines 82 to 94.

Lines 95 to 109: Full of personal opinion statements. In addition: there are results and methodology mixed up in the introduction in an unusual manner and the citations are confusing and in a wrong format.

Material and Methods:
I appreciate the effort of the authors, however this section should be re-written in order to facilitate understanding, please follow examples such as Mahmood et al. (2020) (https://doi.org/10.1038/s41598-020-70406-2).

There are results described in the M&M. For example: Lines 123 to 131, and 139 to 148.

The authors did not perform transcriptome sequencing for the species used in the comparison, correct? Maybe adding a table to summarize the information for this data would facilitate understanding.

Line 153 to 155: Could be explained in the results section. Please re-write to a more scientific sound statement.

Line 170: How did the authors selected “minor isoforms” to be discarded? What were the parameters used?

Lines 262 to 274: Discussion like. Please restrict your information to ensure that the methods are explained with clarity.

Results and Discussion:
Strong statements without literature support (i.e., lines 331-336; 391-397).

For example: “The RBH analysis revealed no transcripts of the lost genes, implying that they were not transferred to the nuclear or the mitochondrial genome, but instead disappeared completely.” Such strong statements should be avoided and substituted by a more conservative and “scientific sound” statement. For example: “Our results suggest that … which corroborates the hypothesis that such genes might not be present in the studied species, such as found by X et al. (20XX) or X et al. (20XX)” (using evidences in other species to support your statement).
• Just to think about: Would it be possible that during your filtering steps some sequences have been removed, or during extraction the transcripts were not sampled given the low quantity of them?

Annotated reviews are not available for download in order to protect the identity of reviewers who chose to remain anonymous.

Reviewer 2 ·

Basic reporting

The paper “Genomic comparison of non-photosynthetic plants from the family Balanophoraceae with their photosynthetic relatives” describe a comparison of transcriptomes features among five species: two non- photosynthetic species Rhopalocnemis phalloides and Balanophora fungosa and their close photosynthetic relatives Daenikera sp., Dendropemon caribaeus, Malania oleifera. The motivation for the study was the unusual features found on the plastid genome of Balanophoraceae family, such as high AT content and substitution rate.
As for the writing, in many cases I consider the text being extremely informal. I suggest the authors to carefully check all text to improve the scientific writing. I will provide some examples in the General comments section.
As for the background provided, figures and tables I consider satisfactory.
I could not find any link to the raw data on the manuscript. This is mandatory for publication on Peer J.

Experimental design

On the general basis the research question was well defined and relevant. The authors provided sufficient information to replicate the analysis, except for the raw data not provided as afore mentioned, and some punctual notes that I raised in the General comments section.

Validity of the findings

Although the subject of the paper is very interesting and relevant for the academia, the authors did not provided accession to the raw data. As the data on which the conclusions are based were not provided the paper did not fit for this criterion.
The detailed points regarding the results and conclusions are in the General comments for the author section.

Additional comments

Introduction

Line 98-109 I consider this text fragment confusing. I suggest it could be rewritten.

Materials and Methods

Line 117 and 132 As a matter of comparison, the plant tissues used for the plant transcriptomes was the same as the present study? Tissue-specific differentially expressed genes were already identified in literature. (See Yu et al 2017- 10.3390/genes8120372 - for example)
Line 123 I couldn’t find the SRA accession numbers for the sequenced Rhopalocnemis phalloides
Line 160 “Bases with sequencing quality less than 3 were trimmed from the 3' end.” I do not understand what “quality 3” really means. Is that a phred score? If so, I suggest that it be clearer.
Line 165 I suggest change the word “removed” to “discarded”
Line 156-166 Is that a reason why the author did not use the command LEADING (trim the reads from the 5’end) for Trimmomatic? I noticed only the TRAILING command, which trims the reads from the 3’end.
Line 184 I suggest change the word “met” to “found”
Line 191-192 “CDSs with proteins that had no matches in NCBI NR, and consequently no definite taxonomic assignment, were kept.” Were the CDSs really kept? Even though they don’t match the NCBI database? I’m not sure if I understood this sentence, could you please check.
Line 215 “DIAMOND as a tool for similarity calculation.” Is that a separated tool used for similarity calculation? Or it is an algorithm incorporated into OrhtoFinder software? Please let this information clearer and inform the citation and software version.
Line 262-274 The information present in those lines are not proper to the Materials and Methods section. I suggest to incorporated then into the Discussion.

Results and discussion
Line 278 I suggest you specify which figure you are referring in the text (Example “The results for all transcripts indicate the near absence of missing genes (Fig 2-A)”.
Line 294 “thus the completeness of the assembly may be overestimated” To ensure the completeness analysis of the assembly I suggest you to align the transcriptome obtained to a transcriptome already available in the literature of a close species and calculate the amount of CDS aligned and the mean percentage of the CDS length that were aligned.
Line 311 “We do not have an obvious explanation for this phenomenon.” The text fragment gives the impression that the authors did not make an extensive search for explanations on the literature. I strongly recommend changing this kind of argument on the manuscript. It does not sound good. A better statement could be such as: “We recommend futhers analysis/studies to enlighten this phenomenon…”.
Line 313 Interesting how? It would be better if you explain why this genes were selected.
Line 331 “The plastid genomes of Rhopalocnemis phalloides and Balanophora fungosa were previously shown to be approximately 10 times shorter than plastid genomes of their close photosynthetic relatives” Is that information on literature? If so, you should quote the paper.
Line 359 “It was logical for us to have a look…” Text fragment consider informal to a scientific writing.
Line 366 “do not differ much from that of their photosynthetic relatives.” Is the numbers statistically different? I think is valid for the authors to test this and incorporated the pvalues on the manuscript.
Line 481 “The increased substitution rate of the nuclear genome (Fig. 3), in our opinion, may be explained in several ways” Another example of text fragment consider informal to a scientific writing.

---

## Round 0.2 · Major Revisions

The revised manuscript presented a significant improvement over the previous version. However, additional points (listed by the reviewer) need to be solved. Furthermore, English is not always handled well, making some sentences difficult to understand. It needs to be gone over by a fluent speaker to clear up these problems. I hope that you will find all advice helpful when revising the manuscript.

Reviewer 3 ·

Basic reporting

English: The English could still be improved throughout the manuscript.

Introduction: The structure of the introduction is sound, and so is the amount of information for the most part. The only thing that is missing is some information on the family Balanophoraceae and the Santalales. The reader might want to learn a bit more on how closely the studies species are. The respective papers should be cited.

l. 40 – “…create organic substances…” – I don’t thing “create” is the right word here, “generate” or “synthesize” might fit better

L. 49 – “The majority of heterotrophic species spend most of the year completely underground…” – There are a few exeptions: Rizanthella garnerii and Hydnora triceps, both flower belowground.

L. 50 - “…since they do not need light for survival.“ – not corrected according to the previous reviewer – it should be stated that they don’t directly depend on light, but indirectly they do through the host (as a previous reviewer suggested).

L. 54 – „…plants have three genomes: the plastid, the mitochondrial and the nuclear.“ – Please rephrase. This is not good language. You mean the plastid genome, the mitochondrial genome, and the nuclear genome. This might be confusing to a native speaker.

L. 55 – „The plastid genome is the smallest and has the highest copy number, thus alleviating sequencing and assembly.“ – Add reference

L. 56 – „These as well as other features of plastid genomes are the reason why most plant genomes sequenced and assembled to date are plastid“ – The conserved structure and the gene density are also reasons why they can be easily assembled. Add respective references.

Section Line 54-64 – There are a lot of relevant references missing.

L. 65 – “The mitochondrial genomes of non-photosynthetic plants are poorly studied…“ – This is a harsh statement and does not acknowledge people who have worked on mitochondrial genomes of non-photosynthetic plants. It is certainly a challenge and hard to get published unless there is one circular mt genome.

L. 69 – „Sequencing of whole nuclear genomes is sometimes expensive due to their large sizes. Therefore, to obtain information about the nuclear genes, scientists often sequence the transcriptome instead of the genome itself.“ – While this might still be true to plants with very large genomes, I still would not state it that way. A proper transcriptome study including various stages and tissues including sufficient replication requires a lot of sequencing as well. Please rephrase. This is again disrespectful to other scientists working on high quality transcriptomes!

L. 75 – „…genes participating in chlorophyll synthesis…“ – To participate implies this is an concious process, but it is not. Please rephrase. See also line 432

L. 76 - „It is hypothesized that the chlorophyll may have some other function apart from photosynthesis“ – Add some examples and references.

L. 87 – „These plastid genomes are about 10 times smaller than..“ – They are a tenth of the size of…

Section L. 85-97 – Please add references! You have to acknowledge other people’s work!!! There are none mentioned in this text!


Literature: There is a lot of literature missing in the introduction. Given the length of the ms, I can’t provide a comprehensive list for the authors.

Structure: The lack of hypothesis early on in the present paper, at the end of the introduction makes it hard for me to understand what the authors are after here. The paper is very descriptive and lengthy up until the last paragraph starting on line 428. Especially the first half of the Results and Discussion section is quite dragging. The structure is clear, but the joint Results and Discussion paragraph is not beneficial to the present manuscript.

Figures : The figures support the data, although they are very minimalistic. The font size in Figure 1 and Figure 3 should be bigger.

Experimental design

Research question:
L. 98 - „Given the unusual features of the plastid genomes of Balanophoraceae, we supposed that their nuclear genomes may also provide information valuable for understanding the evolution of non-photosynthetic plants.“ – This is not a clear hypothesis. The last section of the introduction should contain the hypotheses what is going to be studied and how. It should provide information on what exact data is necessary to answer the specific questions in focus. Most of this is missing here, please work on this section again.

Methods: The RNA for the sequenced species were extracted from very different tissues: Rhopalocnemis phalloides – inflorescence, seedlings, fruits, leaves, and some are unknown. This is not ideal for comparison. I get that the authors took what was available and added Rhopalonemis. So seeing how complete the five transcriptomes are in the BUSCO analysis is quite surprising. However, I wonder if the same analysis using the plant set would reveal in fact genes or gene classes that are missing due to losing the ability to photosynthesize. On the other hand one can only comment on what is present and not what is absent because discrimination between absent and not detected is not possible with the present heterogenous dataset.

Validity of the findings

Overall the analyses are valid for getting a first overview of the situation, which I believe is what the authors went for. When calculatign the dN/dS ratios using over 1,000 genes, it would be interesting to see if there are genes that have accelerated rates and some that have the same rates. More differentiated results would be valuable in the future. Also, I think it would be helpful to include plants outside the Santalaes (like Arabidopsis, where it is possible and makes sense) to be able to pinpoint more genes affected by loss of photosynthesis.

There is a recent study on another parasitic plant Sapria with a similar overall idea to the present ms - Cai, Liming, et al. "Deeply Altered Genome Architecture in the Endoparasitic Flowering Plant Sapria himalayana Griff. (Rafflesiaceae)." Current Biology (2021). The authors take a detailed look at what nuclear genes are present and what they (presumably) do, specific pathways are discussed. This study might be relevant to the present ms.

L. 370 – “The increased AT content of Balanophoraceae is a feature both of non-coding and of coding regions,” – When comparing the AT-content to nuclear genes the AT-content of plastid genes should be used, not the complete plastome (even if it doesen’t make much of a difference)

L. 381 ff – “…it safe to assume that the AT contents of plastid protein-coding genes of Daenikera sp. and Dendropemon caribaeus are somewhere in the range of 60.9%-63.7% or close to that range, because this range is based on 29 plastid genomes of photosynthetic Santalales known to date. – I would leave this out or rephrase this speculative statement. You just never know until you have the data.

L. 451 – “The disappearance of plastid RRR proteins gives an explanation for why..” – change to “possible explaination”

L. 475 – “Analyses of substitution rates in plastids…” – change “plastid” to “plastome”

Additional comments

In the present manuscript with the title “Genomic comparison of non-photosynthetic plants
from the family Balanophoraceae with their photosynthetic relatives” transcriptomic data from the non-photosynthetic plant Rhopalocnemis phalloide was generated and compared to another species from the same family, Balanophoraceae, and three other species from the same order, Santalales. Overall, this is a valuable study. Especially the last paragraph in the Results and Discussion section has some interesting and original ideas. The rest of the text leading up to this could be more concise.

---

## Round 0.3 · Minor Revisions

Although the authors have done a good job revising the manuscript, there are few points that should be fixed, as listed below. I look forward to receiving your revised manuscript.

Comments
-The authors should explain what are the implications of doing transcriptome comparisons when different tissues (e.g., leaves, inflorescences, leaves, fruits) are used.
-Line 628 – Please clarify what N50 means; some readers may be not familiar with this metric.
-In the conclusion section, the authors need to give a clear direction about how to test the hypotheses raised.

---

## Round 0.4 · Minor Revisions

The authors have done a good job of addressing my minor concerns.

However, the transcript assemblies need to be deposited in a public repository before acceptance.

---

## Round 0.5 · accepted · Accept

Thank you for depositing the transcript assemblies of the five plant species in the figshare repository.